# A bio-mimetic miniature drone for real-time audio based short-range tracking

**Roei Zigelman**[1], **Ofri Eitan**[2], **Omer Mazar**[3], **Anthony Weiss**[1], **Yossi Yovel**[2,3,4]*

**1** Electrical Engineering Department, Tel Aviv University, Tel Aviv, Israel, **2** School of Zoology, Faculty of Life Sciences, Tel Aviv University, Tel Aviv, Israel, **3** Sagol School of Neuroscience, Tel Aviv University, Tel Aviv, Israel, **4** School of Mechanical Engineering, The Fleischman Faculty of Engineering, Tel Aviv University, Tel Aviv, Israel

* yossiyovel@gmail.com

## Abstract

One of the most difficult sensorimotor behaviors exhibited by flying animals is the ability to track another flying animal based on its sound emissions. From insects to mammals, animals display this ability in order to localize and track conspecifics, mate or prey. The pursuing individual must overcome multiple non-trivial challenges including the detection of the sounds emitted by the target, matching the input received by its (mostly) two sensors, localizing the direction of the sound target in real time and then pursuing it. All this has to be done rapidly as the target is constantly moving. In this project, we set to mimic this ability using a physical bio-mimetic autonomous drone. We equipped a miniature commercial drone with our in-house 2D sound localization electronic circuit which uses two microphones (mimicking biological ears) to localize sound signals in real-time and steer the drone in the horizontal plane accordingly. We focus on bat signals because bats are known to eavesdrop on conspecifics and follow them, but our approach could be generalized to other biological signals and other man-made signals. Using two different experiments, we show that our fully autonomous aviator can track the position of a moving sound emitting target and pursue it in real-time. Building an actual robotic-agent, forced us to deal with real-life difficulties which also challenge animals. We thus discuss the similarities and differences between our and the biological approach.

## Author summary

Animals solve problems that are considered very difficult for human engineers. In this study, we aimed to mimic animals' ability to localize and track a moving sound source in real time. We do so using a bio-inspired approach by developing a miniature electronic circuit with two ear-like microphones and a micro-processor that is placed on a miniature drone. The circuit detects ultrasonic signals that are typical for echolocating bats and it uses its two 'ears' to estimate the azimuth of the sound source and to steer the drone accordingly. The system is completely autonomous without external human intervention. We focus on bat signals as a proof of concept, but we can alter the electronics to suit other

**Data Availability Statement:** Necessary data and code are available on GitHub: https://github.com/roeizig/thesis-project.

**Funding:** R. Z was partially funded by the Boris Mints Institute. Boris Mints Institute - https://www.

bmiglobalsolutions.org/ and O. E. was partially funded by the Dr Alexander Lester and Eva Lester fellowship. The funders had no role in study design, data collection and analysis, decision to publish, or preparation of the manuscript.

**Competing interests:** The authors have declared that no competing interests exist.

biological signals. Future research will include groups of multiple drones moving together based on acoustic signals as bats and some birds can do in nature.

## Introduction

Many animals are hypothesized to use real-time audio processing to localize and track conspecifics, mates, or prey [1–4]. Some organisms have even evolved special highly accurate ears for this purpose [3]. Especially noteworthy are animals that track sounds emitted by other aviators who are themselves in flight. These animals must apply rapid sensorimotor algorithms in real-time. Mosquitoes, for example, rely on sounds generated by their conspecifics' wingbeat to maintain a cohesive flying group [4,5]. Some bird species recruit conspecifics for collective hunting using vocalizations [6,7] while others migrate at night supposedly using vocalizations to remain in a group [8]. Bats are hypothesized to move in groups in search of prey by eavesdropping on each other's echolocation signals [9–12]. In all these cases, the aviators must detect the desired sounds and then localize them and adapt their own flight-trajectories within dozens of milliseconds. Such animals thus require especially fast sensorimotor algorithms which are both challenging to reveal and potentially beneficial to mimic. Some of the specific challenges include detecting the correct sound signals within background noise, matching the sound signals arriving at the two ears to estimate source-direction and applying relevant algorithms to control the necessary flight maneuver.

Bio-mimetic robotics [13] has become a popular approach to study biological systems. Two of the main goals of this approach include (1) Improvement of current man-made technologies by mimicking animals' abilities to sense and move, and (2) Understanding and testing models of animal behavior with technology that mimics animals' performance while adhering to biological constraints. This second goal has an advantage over theoretical (computer) models, as building an actual device requires solving real-life problems such as overcoming natural external noise, which can be ignored or mitigated when using a computer model.

Various previous studies have applied a bio-mimetic approach to study sonar based movement and localization [14–16]. Others have used this approach to produce bio-inspired sounds localization [17,18]. In this study, we aimed to replicate the ability of certain flying animals to detect, localize and track sounds emitted by another flying aviator in real-time. We equipped a miniature ~30 gr drone (similar in weight to many bat species) with a pair of synchronized ultrasonic microphones and a simple micro-controller, and we developed a fast sensorimotor approach to guide the drone in the direction of a moving platform emitting bat sound signals. In order to keep our approach bio-mimetic, we restricted ourselves to using only two microphones positioned in the same plane with a very short distance between them. Using more microphones or spacing them farther apart would improve localization, but would be less biological. We specifically focused on biological bat echolocation signals, but our approach could be generalized to other signals as well.

In brief, our approach included: (1) an analog high-pass filter circuit that removed most of the low-frequency noise (including the rotor noise) and amplified the bat signals relative to the background noise. (2) Detection of the signals by a simple threshold crossing algorithm. (3) Measuring the time difference of arrival (TDoA) between the two microphones (simulating the ears) and using cross-correlation to assess the direction of the sound source. (4) Steering the drone towards the sound source by turning towards it and maintaining a constant flight speed, controlled by a microcontroller (See Methods for more details).

After developing the system, we performed two experiments to examine its abilities. We show that a rather simple approach can pursue a moving sound source in real-time, and we discuss how our approach differs from that of echolocating bats or other mammals, and how it could be further generalized for both mimetic and technological aims.

## Results

The experiments were conducted in a large anechoic room (5x4.5x2.5m$^3$). In the first experiment, we tested the system's ability to measure the direction of an in-plane (2D) moving sound source, and to turn towards it in real-time without pursuing it (Fig 1A). We played echolocation signals of a *Pipistrellus kuhlii* bat using an Avisoft playback system (UltraSound-Gate 116Hm D/A converter) connected to a Vifa speaker. The speaker was held manually, at a distance of 0.4m from the drone (which was hovering in place). The speaker circled the drone from left to right and vice versa along a 180˚ arc at an angular speed of 15–20 deg/s. We played three bat calls per second at an intensity of 105dB SPL (measured at 1m)–mimicking a bat during search phase when the intervals between the calls are relatively long [9,19,20]. The system filtered out the low-frequency noise, including the noise generated by the rotors (which has substantial energy at frequencies of up to 25KHz), and detected the bat-signals at a very high rate from a distance of up to 0.8m (Fig 1B). We used video tracking to monitor the position and heading of the speaker and the drone in all experiments (see Methods). The drone successfully tracked the direction of the sound source, maintaining an angle of 34 ° ± 27° (Mean ± SD) relative to the sound source and under 20 ° during 35% of the time (Fig 1C and 1D and S1 Movie).

The second experiment tested the ultimate goal of the system, that is, to pursue a moving target emitting ultrasonic signals. Here, too, the playback speaker was hand-held (while playing bat signals as above) and moved around at an average speed of up to 0.85 m/s. The experimenter moved the speaker in various trajectories including spherical, figure-eight and random

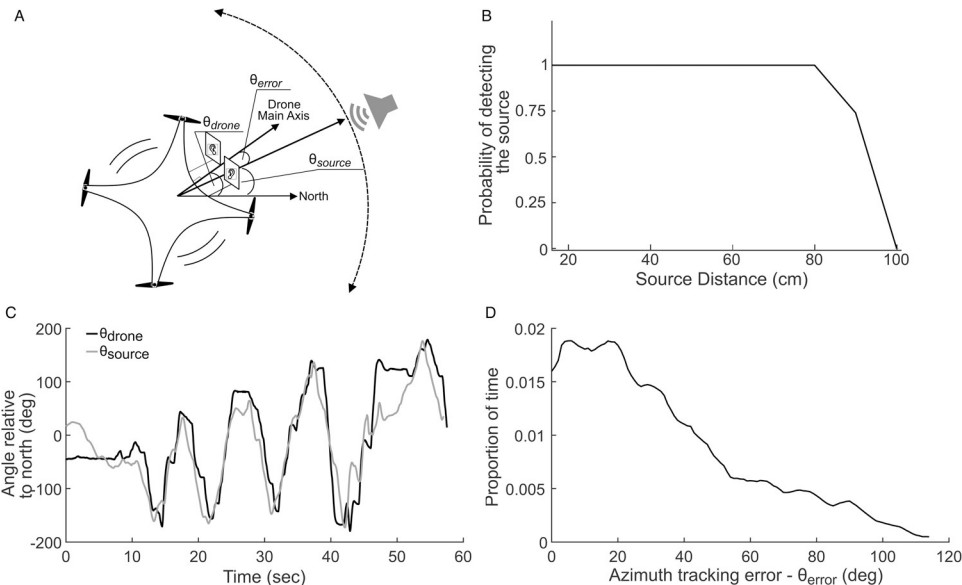

**Fig 1. The drone tracked the direction of a sound source in real time.** (a) Schematic of the experiment. (b) The probability of detecting our played-back sound signals. A hundred signals were emitted for each distance and their detection rate was computed. (c) The heading angle of the drone (black) and the sound source (grey) relative to north for one example trial. (d) Overall orientation error histogram.

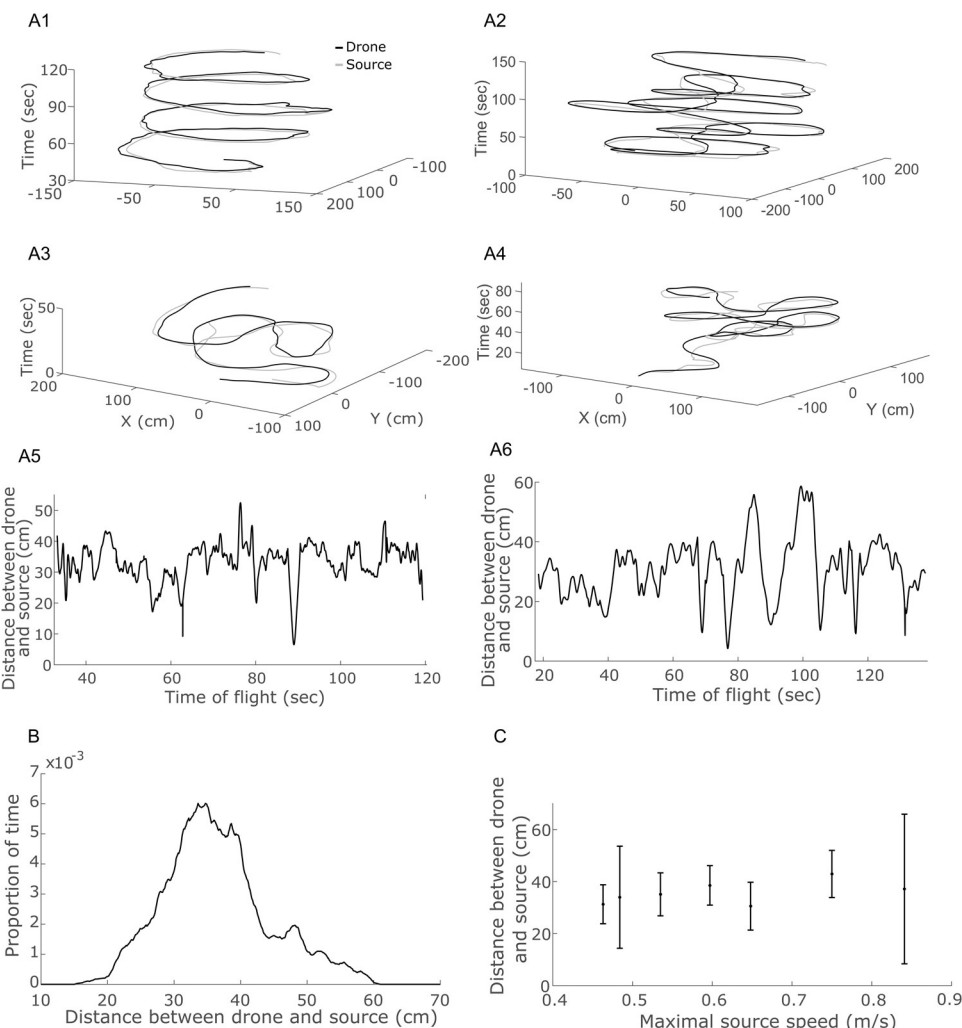

**Fig 2. Our mimetic-drone was able to track a sound-emitting moving target.** (a) Four examples of tracking trials, in a 2-Dimensional plane. Black line depicts the drone's movement and grey is the target (i.e., the speaker). The z-axis shows time. Examples from top left to bottom right include–(a1) circular movement, (a2-3) two figure eight trajectories, (a4) and one random trajectory respectively. (a5-6) Two examples of the error (i.e., the distance between drone and source) over time during circular movement (trial a1) and figure eight (trial a2) respectively. (b) The overall, for all trials, pursuit distance histogram. (c) The drone's tracking performance was independent of speed for the range of tested speeds.

movements. In all cases, the speed and the exact trajectory were inherently noisy due to the difficulty to maintain an accurate speed when moving the target manually. In all trials, the drone successfully tracked the target, managing to remain in close proximity to the target (the mean distance was 37 ± 8cm (Mean ± SD), see Fig 2 and S2 Movie). The drone's tracking error, the distance between the robot and the source of transmission, was independent of the velocity of the source, in the range of velocities tested (0.18–0.28 m/s, there was no correlation between error and speed, Pearson correlation, R = -0.372, P = 0.412, N = 7).

## Discussion

This work demonstrates that our approach, which relied on real-time processing using rather simple algorithms, was enough for tracking a slowly moving sound source. Our sensory

processing included a high-pass filter, a threshold detector, and finding the peak of the cross-correlation between the two channels. The target in our experiments was moving rather slowly (more similar to an insect than a vertebrate) but it is likely that our system could be upgraded to allow tracking faster targets, without any dramatic changes to the algorithm. First, the microcontroller performed a cross-correlation between the signals arriving at the two microphones in windows of 28ms. This imposed a rather long integration time, and it dictated a slow response time. We could elevate the system's response speed by using shorter integration windows. Notably, our target was also rather slow in its emission rate—emitting only 3 calls per second, less than a typical bat would do when flying with nearby foraging bats. The emission rate of the target will determine the maximum potential sensory update rate and will thus affect the response time of the tracker.

Our approach was oversimplified in several ways that could be improved in the future. To simplify the analysis and due to the lack of a pinna-like model, we restricted the tracking to two dimensions. Notably, although bats move in 3D [21], their movement in the third dimension is much lesser in comparison to that in the horizontal plane when foraging in a restricted space. Our approach could be generalized to 3D by adding pinnae-like structures on-top of the microphones. As is well documented in mammals [22,23], such pinnae provide elevation-specific filtering which complete the time of arrival differences and provide 3D information. As observed in real bats such pinnae could be small and light-weight and could thus be carried even by our miniature drone.

Moreover, the approach presented in this study only dealt with a single sound source and thus avoided the problem that often occurs in reality when more than a single sound source has to be tracked or when multiple sources have to be separated. A future algorithm will deal with such situations, localizing more than a single source and either selecting one of them or following some weighted average of their directions. In a previous work of our group, it was shown how multiple sources can be localized within a single sound-beam [24]. In our current trials, the acoustic target was hand-held and moved by a human experimenter who might have biased the movement to ease tracking (but also added stochasticity to the movement–see Fig 2). In future experiments, we aim to extend our approach to several drones autonomously moving while tracking each other according to a commonly simulated agent based model for collective behavior [25].

We did not use active sonar as we have done previously in a terrestrial system [24], but only mimicked passive hearing localization abilities. In this sense, our model is reduced in comparison to actual bats, but a future drone can hopefully integrate our passive tracking algorithm with active sonar. These conditions made the task easier for the system and it would be interesting (and difficult) to test it under more realistic multiple bat-situations.

Finally, in this work, we focused on tracking bat signals, but our approach could be easily generalized to other signals as well.

## Methods

Overall, the system consisted of an in-house developed PCB which included two microphones (Knowles, SPU0410LR5H), a (High Pass Filter) HPF circuit, 2 ADCs (Analog Digital Converters) with DMA (Direct Memory Access), a STM32F446 (ST) micro-controller and a UART Rx/Tx chip for communication with the drone. This system was installed on a Crazyflie 2.0 platform (by Bitcraze AB). The entire system can be seen in Fig 3A. The PCB is $28x20$ mm$^2$ large and weighs 1.4 gr, making the total weight of the drone 34 gr (similar to many bat species). The drone itself, with no additions, weighs 27 gr and has a maximal takeoff weight of 42 gr.

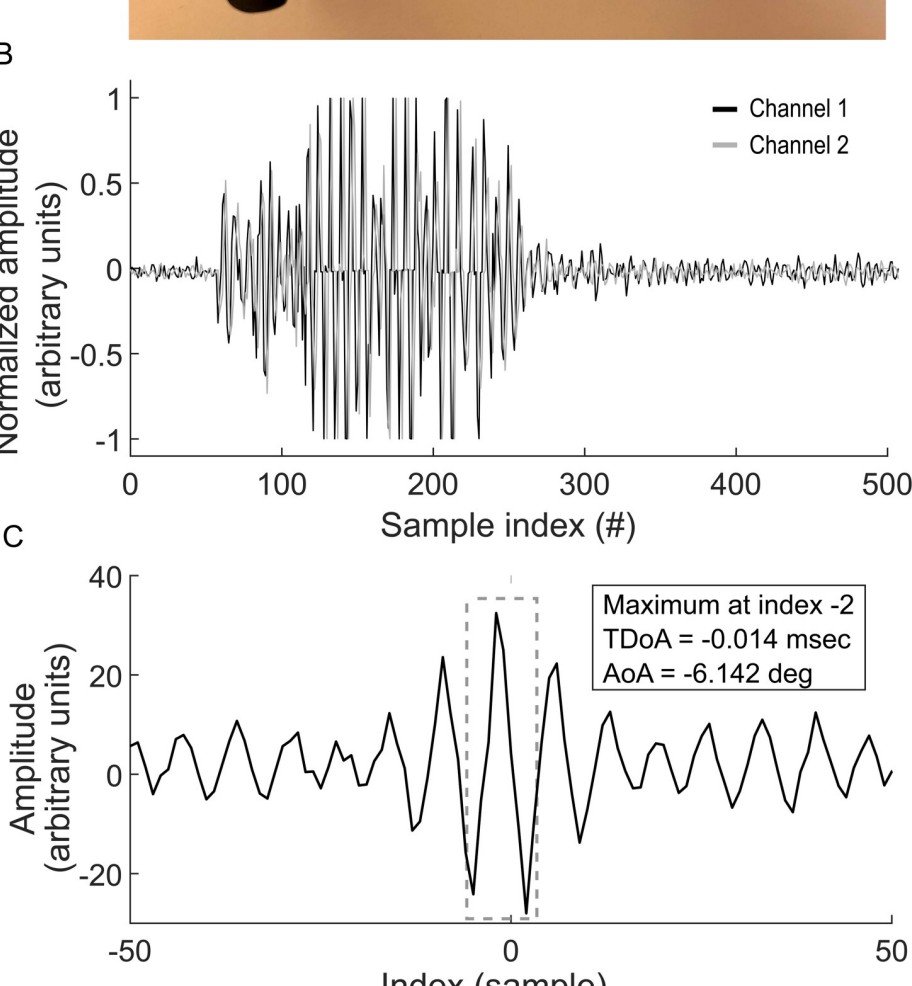

**Fig 3. The drone found the angle to the source of transmission.** (a) Image of the entire system–the drone with the embedded electronics. (b) Recording of the bat chirp, as received by the system's microphones. (c) Cross-correlation between the recordings from both microphones, peak-detection and angle-of-arrival calculation, as done by the PCB integrated microcontroller.

The two synchronized microphones, acting as the system's ears were mounted on the drone spaced 5cm apart. The Knowles microphones are almost omnidirectional and they are sensitive in the range between 10Hz to 100kHz, but their frequency response is highly nonlinear (link). The microphones input was filtered with a designed amplification circuit which helped compensating for this frequency response. The receiving circuit included a second-order HPF with a cutoff frequency of 48kHz, a damping factor of 0.5, and gain of 6dB. The system allowed recording of the bat signals after removing much of the noise at the lower frequencies. The signal was recorded by the STM32F446 micro-controller at a sampling rate of 144 kHz in each in windows of ~28 ms. The micro-controller cross-correlated the two recordings and performed a peak detection algorithm on the resulting signal (Fig 3B and 3C) to estimate the time difference between the two according to:

$$\tau_{difference} = \frac{arg\ max\ (f*g)[n]}{f_{sampling}}, n \in [-a, a]$$

where $\tau_{difference}$ is the time difference of arrival between the two microphones, $f$ and $g$ are the sets of samples from each microphone, each with a length of $2^*a$ samples, $[-a,a]$ is the closed interval domain of the cross-correlation function between $f$ and $g$, $n$ is the independent discrete variable and $f_{sampling}$ is the sampling frequency. Assuming the distance from the source of transmission is much greater than the distance between the microphones, a planar wave assumption can be made regarding the propagation of sound in air. This assumption allows the calculation of the angle of arrival (AoA) $\theta$:

$$\theta_{arrival} = \arccos\left[c_{air} \cdot \frac{\tau_{difference}}{d_{mics}}\right]$$

Controlling the drone: Once a threshold of 0.65V above the microphones' DC (Direct Current) offset was crossed in the ADC GPIOs (General Purpose Input/Output) (Fig 3) the microcontroller started recording the microphones, estimated the azimuth of the sound source and steered the drone accordingly. The drone is equipped with a microcontroller running embedded software tailored for its Real-Time Operating System (RTOS). Our sensory microcontroller thus transmitted the estimated turning angle to the drone which in response rotated by a degree of $\theta_{arrival}$ and at an angular velocity of up to 3 rad/s while remaining stationary in experiment 1 or while moving forward in which a constant speed of 0.3m/s in experiment 2. After reaching the required angle, the drone kept its direction of flight with constant velocity, till the next angle command from the microcontroller. The drone maintained a constant height above ground using an optic sensor.

## Tracking the experiments

The sound-source (the playback speaker) and the drone were tracked using the motion analysis tracking system composed by 20 tracking cameras (16 Raptor E 1280 × 1024 pixels cameras, and four Raptor-12 4096 × 3072 pixels cameras, Motion-Analysis Corp.). Motion was tracked at 200 fps and with a spatial resolution of less than 1 mm (see full details regarding the tracking accuracy in supplementary Fig 1 of ref [26]). To enable tracking, 6 mm spherical reflective facial markers (3X3 Designs Corp.) were glued to the drone and speaker using a double-sided tape. Of the 20 trials we ran only eight contained usable data. In the other trials there was no data for various technical reasons: either the tracking system did not start, the drone's battery finished or the drone exited the FOV covered by the system, etc. Seven of these eight trials were fully analyzed.

During the first experiment (the drone didn't fly towards the target) the detection probability of the sensory system was measured. The source of transmission was aimed at the

microphones, and 100 chirps were transmitted at each distance. The motors were operating at full thrust for the purpose of maximal noise injection into the microphone recordings.

The detection probability was calculated by:

$$p(detection|distance) = \frac{amount\ of\ chirps\ detected}{amount\ of\ chirps\ transmitted}$$

## Supporting information

**S1 Movie. Movie presenting the first experiment where the drone turns towards the direction of the sound source (a speaker held by the experimenter).**
(MP4)

**S2 Movie. Movie presenting the second experiment where the drone autonomously tracks the sound source (a speaker held by the experimenter).**
(MP4)

## Author Contributions

**Conceptualization:** Ofri Eitan, Yossi Yovel.

**Data curation:** Ofri Eitan, Omer Mazar.

**Formal analysis:** Roei Zigelman, Ofri Eitan, Omer Mazar.

**Methodology:** Anthony Weiss, Yossi Yovel.

**Software:** Roei Zigelman.

**Supervision:** Anthony Weiss, Yossi Yovel.

**Visualization:** Roei Zigelman.

**Writing – original draft:** Roei Zigelman, Yossi Yovel.

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
