## [Decision Letter · Decision Letter 0]

28 Nov 2021

Dear Mr Zigelman,

Thank you very much for submitting your manuscript "A bio-mimetic miniature drone for real-time audio based tracking" for consideration at PLOS Computational Biology. As with all papers reviewed by the journal, your manuscript was reviewed by members of the editorial board and by several independent reviewers. The reviewers appreciated the attention to an important topic. Based on the reviews, we are likely to accept this manuscript for publication, providing that you modify the manuscript according to the review recommendations.

Specifically the title could better reflect the locality of the tracking capability.

In addition it would be helpful to the reader to state clearly, early on, that sound localization is implemented in 2D.

I also share a concern of a reviewer that “Of the 20 trials that were recorded the seven that had most consistent video tracking were analyzed.” - this looks dubious unless there is sufficient justification for selecting these data. It seems unlikely that such a tracking system would completely fail to track, so even if trajectories are interrupted it will be very helpful to include/show these data.

Further comments and changes are listed below in the referee reports. 

Sincerely,

Iain Couzin

Guest Editor

PLOS Computational Biology

Natalia Komarova

Deputy Editor

PLOS Computational Biology

[LINK]

Reviewer's Responses to Questions

**Comments to the Authors:**

Reviewer #1: In the manuscript titled “A bio-mimetic miniature drone for real-time audio based tracking” the authors report a study on a performance of a small autonomous flying robot that uses two microphones, as well as a custom-developed microcontroller, to locate the orientation of a sound emitter. By orienting towards the sound source, the drone could follow a single speaker – given that the motion speed of the source is small enough so that the drone does not lose it, and its velocity does not change very abruptly. The authors discuss the performance and the limitation of their approach to some extent. The manuscript is generally well written, it is clear and easy to follow.

This is a very exciting line of research, in my opinion, but I have several crucial concerns, which honestly makes it hard for me to give a clear recommendation.

Major concerns:

I am not convinced that the topic of the manuscript fits well in the field of Computational Biology. Although it’s connection to biology in its aims are obvious, the manuscript does not discuss any biological implications in detail. It is also not clear whether the actual implementation could or could not be used to study animals. The manuscript may better suit in a more technological journal?

The title is not specific enough. It does not specify that the method could only work for locating a single sound source and within a very short range. I am sceptical about how the current implementation could overcome these limitations, so at least the title should clearly state them somehow.

The method they use and report in this manuscript works for sound localisation only in 2D. Although the authors point this out in their discussion (2nd paragraph), but I think it is a major limitation that should be made clear from the beginning. Generalizing the system to work in 3D (using only 2 microphones) would be a hard challenge.

Minor comments:

Fig. 1c. The box for the keys is misaligned. Using “drone” and “source” here is a much better choice than “Crazyflie” and “Speaker” as on Fig. 2. This inconsistency should be resolved.

Page 8, last line: N should be given for the Pearson correlation.

Page 9, Fig. 2. Legend, last line: “Of the 20 trials that were recorded the seven that had most consistent video tracking were analyzed.” This sounds worrying, as it is not clear how the presented data was selected. At page 11, the Authors state about the tracking: “Motion was tracked at 200 fps and with a spatial resolution of less than 1 mm.” So either there they should report any other source of tracking error or proportion of missing track points, or they should report results calculated for an entire recorded data set.

Some of the abbreviations are not introduced.

In the 3rd paragraph of the discussion, the Authors refer to an unpublished work (Ref. 24. Elyakim, I., Kosa, G. & Yovel, Y. An autonomous navigating and mapping acoustic robat. Under review. ). Or is this meant to be a reference to Elyakim, I., Cohen, Z., Kosa, G. & Yovel, Y. A fully autonomous terrestrial bat-like acoustic robot, PLOS Computational Biology, 14.9 (2018)?

Page 11, equations: The two equations should match in their appearance. If tau_difference is defined in the first equation, I see no reason why this is not used in the second equation.

I think the figure titles could be improved by changing them to summarise what is illustrated on the figure, and not just state what the drone did.

Typos:

Fig. 2. Legend, line 2. “in2-Dimentional”

Summary

As a consequence of all the above, I can’t recommend publication in PLOS Computational Biology.

Reviewer #2: This paper describes a biomimetic robot inspired by bats that uses two microphones and a time difference of arrival approach to track and turn toward a speaker producing high-frequency pings. Two experiments are performed to demonstrate the capabilities of the robot - one where the robot remains stationary and turns toward the speaker, and one where it moves with a constant velocity and pursues the speaker. The article is clearly written and the results demonstrates proof-of-principle that the system can perform this tracking task reasonably well. As I am not an expert in robotics or sound localisation, I cannot assess the novelty of the approach, however I found the paper interesting and the results seem sound. I have a few suggestions for improvement.

1. The authors provide a link to a Github repository containing the source code for the project. I am glad to the code is available, however there is very little documentation, which would make it difficult for someone to reproduce. I think this issue is especially relevant as the paper is really about demonstrating a method as a proof of principle, thus a large part of the value would be in allowing other researchers to use and expand on the system. I would therefore suggest better documentation to allow potential replication of the setup.

2. It is mentioned in the manuscript that a tracking system using 24 cameras was used to track the drone and playback speaker. Is this system described elsewhere? If so, it would be good to provide a reference, and if not, I think more detail should be included on the tracking system.

3. The tests have been carried out by a human manually moving a speaker around. This probably introduces some bias into the measurements, as the human (who probably wants the robot to succeed) may subtly respond to the robot by slightly modifying their behavior, e.g. to slightly wait for it if it is behind. The ideal case for testing would have been to have the speaker movement automated, however I recognise this might have been more trouble than it is worth. Overall, I don’t see this as a huge issue as the paper mainly serves as a proof of concept, but it might be worth just acknowledging this as a potential limitation.

Line edits:

As a general comment, it would be nice to provide line numbers to simplify review. I have done my best here to describe where these edits are located.

Pg 1 paragraph 3: “theoretic” should probably be “theoretical”

Pg 2 Results: “hovering at place” —> “hovering in place”

Pg 6 “the system allowed to record” —> “the system allowed recording of” or “the system allowed us to record”

Figure 2: I suggest labelling the legend as “drone” rather than “crazyflie” for clarity

**Have the authors made all data and (if applicable) computational code underlying the findings in their manuscript fully available?**

Reviewer #1: Yes

Reviewer #2: Yes

PLOS authors have the option to publish the peer review history of their article (what does this mean?). If published, this will include your full peer review and any attached files.

Reviewer #1: No

Reviewer #2: No

Figure Files:

Data Requirements:

Reproducibility:

References:

---

## [Editor Report · Decision Letter 1]

18 Feb 2022

Dear Dr. Yovel,

We are pleased to inform you that your manuscript 'A bio-mimetic miniature drone for real-time audio based short-range tracking' has been provisionally accepted for publication in PLOS Computational Biology.

Before your manuscript can be formally accepted you will need to complete some formatting changes, which you will receive in a follow up email. A member of our team will be in touch with a set of requests. In addition, I would suggest that you consider revising the title. Currently it is rather cumbersome, especially since "audio based" should be "audio-based", resulting in many hyphenated terms. Perhaps you could employ a simpler alternative that reads more clearly, such as "A bio-mimetic miniature drone for real-time, local acoustic tracking" or "A bio-mimetic miniature drone for real-time tracking of a local sound-emitting target".

Best regards,

Iain Couzin

Guest Editor

PLOS Computational Biology

Natalia Komarova

Deputy Editor

PLOS Computational Biology

---

## [Editor Report · Acceptance letter]

3 Mar 2022

PCOMPBIOL-D-21-01562R1 

A bio-mimetic miniature drone for real-time audio based short-range tracking

Dear Dr Yovel,

I am pleased to inform you that your manuscript has been formally accepted for publication in PLOS Computational Biology. Your manuscript is now with our production department and you will be notified of the publication date in due course.

With kind regards,

Zsofia Freund
